# Full Arch Implant-Prosthetic Rehabilitation in Patients with Type I Diabetes Mellitus: Retrospective Clinical Study with 10 Year Follow-Up

**DOI:** 10.3390/ijerph191811735

**Published:** 2022-09-17

**Authors:** Bianca D’Orto, Elisabetta Polizzi, Matteo Nagni, Giulia Tetè, Paolo Capparè

**Affiliations:** 1Dental School, Vita-Salute San Raffaele University, 20132 Milan, Italy; 2Department of Dentistry, IRCCS San Raffaele Hospital, 20132 Milan, Italy; 3Center for Oral Hygiene and Prevention, Dental School, Vita-Salute San Raffaele University, 20132 Milan, Italy

**Keywords:** systemic diseases, edentulism, dental implants, diabetes mellitus

## Abstract

The aim of this retrospective clinical study was to evaluate and compare implant survival, marginal bone loss, and clinical and prosthetic complications in healthy patients and patients with type I diabetes undergoing full arch implant-prosthetic rehabilitation. A total of 47 patients needing total fixed rehabilitation of one or both arches were enrolled for this study. Based on the absence of any systemic diseases (Group A) or the presence of type I diabetes (Group B), the sample was divided into two groups. According to the grade of bone atrophy in the posterior region, patients received full arch rehabilitation (of one or both jaws) with 6 axial implants or, if the residual posterior bone height was insufficient, All-on-Four rehabilitation and a total 236 dental implants were placed. Follow-up visits were performed 1 week after surgery, at 3 and 6 months and then once a year for the next 10 years. No statistically significant differences between groups were recorded about implant survival rates, marginal bone loss, or clinical and prosthetic complications. However, concerning complications, post-surgical bleeding and wound infection were recorded in Group A more than in Group B. In cases of compensated diabetes compensation, implant placement could be considered a safe procedure.

## 1. Introduction

As the average age increases, edentulism may represent an increasingly frequent problem [1].

Dental implants in support of fixed prostheses could represent the most appropriate therapeutic choice in the rehabilitation of the edentulous patient [2].

The loss of teeth results in significant bone resorption, especially in the posterior region, often hindering the placement of traditional implants [3,4].

If the amount of bone in the posterior sectors is sufficient, the placement of six axial implants to support fixed total dentures may be preferred [5,6]; in contrast, rehabilitations with a reduced number of implants based on the insertion of tilted fixtures to exploit the basal bone could represent a valid therapeutic alternative to avoid more invasive procedures with a higher number of complications [7].

In this perspective, the “All-on-Four” method, involving immediate loading, could reduce the intra- and post-operative risks of bone grafting and sinus lift, decrease the clinical time required, and promote patient acceptance [8,9].

The increase in the average age of the population, however, coincides with an increased incidence of several systemic diseases, including diabetes, defined by the ADA (American Diabetes Association) as a group of metabolic diseases characterized by elevated blood glucose levels (hyperglycemia) resulting from the body’s inability to produce or employ insulin [10].

As reported by several authors, this pathology affects both the oral microbiome, vascularization and healing processes, and salivary composition [11,12,13].

In cases of implant rehabilitation, the incorrect management of the diabetic patient may increase the risk of both intra-operative complications, such as hypoglycemic crisis [14], and post-operative complications, such as failure of osseointegration, impaired soft tissue healing, mucositis, and peri-implantitis [15,16,17].

The aim of this retrospective clinical study was to evaluate and compare implant survival, marginal bone loss, and clinical and prosthetic complications in healthy patients and patients with type I diabetes to assess whether there could be differences between healthy and type I diabetes compensated patients undergoing full arch implant-prosthetic rehabilitation.

## 2. Materials and Methods

### 2.1. Patients’ Selection

This comparative retrospective clinical study was performed at the Department of Dentistry, IRCCS San Raffaele Hospital, Milan, Italy. The ethics committee approval number is 190/INT/2021.

The study was conducted in accordance with the tenets of the Declaration of Helsinki and followed the Strengthening the Reporting of Observational Studies in Epidemiology (STROBE) guidelines for cohort studies (http://www.strobe-statement.org, accessed on 24 April 2021). During the period from January 2011 to November 2021, patients with total edentulous maxilla and/or mandible or severe impairment of residual teeth in one or both jaws were consecutively enrolled.

The eligibility criteria were as follows: edentulism of one or both arches or severe impairment of residual teeth, requiring fixed prosthetic rehabilitation, the absence of any systemic diseases, or the presence of diabetes as single pathology.

Exclusion criteria were smokers, the presence of other systemic diseases than diabetes, uncompensated diabetes, bisphosphonates therapy, head and neck radiotherapy in less than one-year, severe malocclusion, severe parafunction, and the inability to adhere to home and professional hygiene maintenance protocols.

All diagnoses were made clinically and radiographically. The radiographic examination was conducted at first level with panoramic radiography and at second level with cone beam computed tomography (CBCT) to identify residual bone height and width.

Depending on the absence of any systemic diseases (Group A) or the presence of type I diabetes (Group B), the sample was divided in two groups.

Blood tests were prescribed about one month before surgery to check patient’s general health and to identify possible signs of uncompensated diabetes [18].

### 2.2. Implant-Prosthetic Protocol

Antibiotic prophylaxis (2 g amoxicillin 1 h before surgery) was only given to patients affected by diabetes mellitus [19].

Surgery was performed under anesthesia induced by local infiltrations of opticain 120 solution with adrenaline 1:80.000 (AstraZeneca, Milan, Italy).

A crestal incision, made in a palatal or lingual direction to obtain more keratinized tissue, and medial and distal vertical release incisions allowed the creation of a full-thickness flap. The exposed bone ridge was regularized with a straight handpiece and an oxivore forceps. Midline, maxillary sinus region, and mental nerve were identified with a sterile pencil as reference sites for the fixtures’ placement.

According to residual bone height in posterior jaws [20], viewed from CBCT [21], six straight implants or two straight and two mesially and distally tilted implants were positioned. A lanceolate drill was employed to perforate cortical bone. A pilot drill of ø 2.00 was employed to create an implant way insertion and to define fixture’s setting. A positioning pin was plugged to verify implant location, emergence and, when it occurred, angulation. Progressive diameter drills were applied up to the final fixture’s diameter. The site was over-prepared vertically and sub-prepared transversely to promote the primary mechanical stability. The insertion torque ranged between 30 and 40 N·cm before final seating of the implant, allowing for the immediate loading. A manual screwier was applied when incomplete seating of the implant occurred.

When six axial fixtures were placed, straight or angulated abutments at 17 degrees were applied to compensate for any lack of parallelism.

In cases of “All-on-Four” protocol, angulated abutments at 30 or 45 degrees were screwed on tilted implants and straight on axial.

Flap adaptation and suturing was performed around closure screws or extreme abutment with 3-0 non-resorbable sutures (Vicryl; Ethicon, Johnson & Johnson, New Brunswick, NJ, USA).

Titanium cylinders were screwed to abutments; an all-acrilyc resin provisional prothesis, obtained from preliminary impressions taken one week before surgery, was drilled in their correspondence to take pickup impressions (Permadyne, ESPE, Seefeld, Germany).

Intra-oral x-rays were performed to assess the correct implant positioning. Antibiotic therapy (amoxicillin 1 g twice daily for 6 days after surgery) and analgesic therapy (non-steroidal anti-inflammatory drugs, as needed) were prescribed for each patient. Mouth rinsing with a chlorhexidine digluconate-containing solution (0.12% or 0.2%), were recommended twice daily for 10 days. One week after surgical procedure, provisional prothesis was unscrewed to remove sutures.

About 3 h after the surgery, a screw-retained, metal reinforced acrylic provisional prosthesis of a maximum 12 teeth and without cantilever, was delivered. Screw access holes were covered with provisional resin (Fermit, Ivoclar Vivadent, Naturno, Bolzano, Italy).

Four months later, the provisional prothesis was replaced with an acrylic resin implant-supported definitive prostheses with titanium framework and provided with distal cantilever.

Articulating paper (Bausch, Nashua, NH, USA) was applied to check the occlusion, which reproduced the natural dentition.

The screw access holes were covered with acrylic resin (Fermit, Ivoclar Vivadent Naturno, Bolzano, Italy).

### 2.3. Follow-Up

Follow-up visits were performed 1 week after surgery, at 3 and 6 months and then once a year for the next 10 years.

Professional oral hygiene sessions were carried out every 4 months after surgical-prosthetic procedure.

### 2.4. Clinical Outcomes

#### 2.4.1. Implants Survival Rate

The implant survival rate was based on implant loss during the follow-up period.

After evaluating the implant survival rate in each group, Group A and Group B were compared to determine whether there were differences between healthy patients and patients with type I diabetes.

#### 2.4.2. Marginal Bone Loss

Intra-oral x-rays were taken at 3, 6, 12 months and once a year during the follow-up period. The software, (DIGORA 2.5, Soredex, Tuusula, Finland), was calibrated for each image using the known diameter of the fixture at the most coronal portion of the implant neck to assess marginal bone differences. The linear distance between the most coronal point of the bone-implant contacts and the coronal margin of the implant neck was measured on the mesial and distal sides, to the nearest 0.01 mm, and then the average was calculated. Bone level changes of individual implants were averaged at the patient level and then at the group level. The results were statistically evaluated and compared.

#### 2.4.3. Clinical Complications

Possible clinical complications such as post-surgical edema, pain while taking analgesics drugs, bleeding and/or wound infection were recorded during follow-up checks.

#### 2.4.4. Prosthetic Complications

Any prosthetic complication such as provisional prosthesis fracture, provisional screw loosening (abutment) provisional screw loosening (prosthetic) and/or detachment of the veneering material (final prosthesis) was signed during patients’ visit.

### 2.5. Statistical Analysis

Statistical analysis was carried out using Python 3.8.5 and the following packages: math, sciPy and pandas. According to the sample distribution, variance, and experimental setting, we used parametric independent samples t-test, Pearson’s chi-quare test, or z-test to test for/against differences between groups. Across all analyses, *p*-values < 0.05 were considered significant. Data were analyzed at the aggregate level.

To investigate differences in terms of implant survival rates, clinical complications, and prosthetic complications between Group A and Group B, Pearson’s chi-quare and z-tests were applied at a significance level of *p* < 0.05.

To compare marginal bone loss between Group A and Group B at 6 months and annually until the 10-year follow-up, Pearson’s chi-square and Student’s t-tests were applied at a significance level of *p* < 0.05.

## 3. Results

According to inclusion and exclusion criteria, 47 patients (26 females, 21 males) with total edentulism of one or both arches or need for avulsion of residual impaired teeth were enrolled for this study. The mean age was 68 years (range: 57–79). The sample was divided in two groups: 25 patients were included in Group A, 22 in Group B.

According to the grade of bone atrophy in the posterior region, patients received full arch rehabilitation (of one or both jaws) with 6 axial implants or, if the residual posterior bone height was insufficient, All-on-Four rehabilitation and a total 236 dental implants were placed (Table 1).

### 3.1. Implants Survival Rate

About implant survival rates, in Group A, four implants were lost during the follow-up, in Group B, six (Table 2).

No statistically significant differences in implant survival rates (early and late failures rates) between Group A and Group B were observed (*p* > 0.05). The differences between the two groups at a 95% confidence level appear not to be significant enough to reject the null hypothesis, and the two groups should be considered statistically not different.

### 3.2. Marginal Bone Loss

Concerning marginal bone loss, values recorded were summarized according to measures obtained during the follow-up and the average for each group (Table 3).

No statistically significant differences in marginal bone loss between Group A and Group B were observed at any of the follow-up evaluations (*p* > 0.05). The differences between the two groups at a 95% confidence level appear not to be significant enough to reject the null hypothesis, and the two groups should be considered statistically not different.

### 3.3. Clinical Complications

About clinical complications (Table 4), post-surgical edema was recorded in two cases in Group A and four in Group B. No patients reported post-surgery pain. Post-surgical bleeding and wound infection were recorded in patients affected by type I diabetes more than patients of Group A.

Except for the variable bleeding, the differences in terms of clinical complications (edema, pain, bleeding, and wound infection) between the two groups at a 95% confidence level appear not to be significant enough to reject the null hypothesis. The results of Pearson’s chi-square test and the z-test at a 99% confidence level suggest that there is not a statistically significant difference in clinical complication between Group A and Group B.

### 3.4. Prosthetic Complications

Concerning prosthetic complications (Table 5), provisional prothesis fracture occurs in two cases in Group A, in two in Group B; provisional screw loosening (abutment) was recorded in three cases in Group A and three in Group B; in only one case provisional screw loosening (prosthetic) occurred. No detachment of the veneering material (final prosthesis) was recorded during follow-up.

No statistically significant differences in prosthetic complications (provisional prosthesis fracture, provisional screw loosening (abutment), provisional screw loosening (prosthetic), and detachment of the veneering material (final prosthesis)) between Group A and Group B was observed (*p* > 0.05). The differences between the two groups at a 95% confidence level appear not to be significant enough to reject the null hypothesis, and the two groups should be considered statistically not different.

This section may be divided by subheadings. It should provide a concise and precise description of the experimental results, their interpretation, as well as the experimental conclusions that can be drawn.

## 4. Discussion

Patients selected for this study underwent an anamnestic questionnaire and, in prevision of implant surgery, blood tests were required to assess their general health status and adequacy for the planned surgery.

Systemic evaluation of the patient undergoing implant surgery by means of proper collection of anamnestic data and adequate evaluation of blood tests could be crucial [18,19].

Uncompensated systemic pathologies could interfere with implant osseointegration, implant survival rate, and promote intra- and post-operative complications [20].

In diabetic patients, as reported by de Bedout et al. 2018 [21], the threshold value of fasting blood glucose level to safely perform oral surgery procedures is 180 mg/dL or 200 mg/dL; in cases of higher values but less than 234 mg/dL, emergency surgical procedures could be performed, while any dental treatment should be avoided if fasting blood glucose level is equal to or greater than 240 mg/dL [22,23].

Before undergoing the surgical procedure, all patients were administered antibiotic prophylaxis.

As suggested by Ramu et al. [24] and confirmed by other studies [25,26], the administration of antibiotic prophylaxis in the diabetic patient could be considered mandatory in the case of uncompensated pathology but is also recommended in other cases.

Implant survival in the present study was 96.72% in healthy patients and 94.74% in diabetic patients.

No statistically significant differences were found between the groups in terms of either implant survival or marginal bone loss.

Similar results were obtained by Sannino et al. in their study on the rehabilitation of the edentulous posterior maxilla in which they compared healthy patients and patients with compensated type I diabetes, in which no statistically significant differences in either parameter was recorded [27].

Sghaireen et al. in their retrospective clinical study at the 3-year follow-up compared implant failure in healthy patients and patients with compensated diabetes, and a non-significant (*p* = 0.422) failure rate was found in the case group in comparison to the control group (9.04%), concluding that, if diabetes is compensated, there are no additional risks compared to a healthy patient undergoing dental implant therapy [28].

Regarding the impact of diabetes on marginal bone loss, the results of the literature may be conflicting.

In contrast to the present study, Souto-Maior et al. in their systematic review found that, although no differences in implant survival were found between healthy and diabetic subjects, diabetes negatively influences marginal bone loss [29].

Similar results were reported by Tan et al. [30] in a previous study, in which they attributed marginal bone loss to an uncontrolled glycosylated hemoglobin (HbA1C) value, which as reported by Diz et al. is also associated with an increased risk of post-operative complications [31].

The role of HbA1C was also confirmed by Robertson et al. in their literature review, stating that surgical procedures can only be performed safely if the value of this protein is below 7% [32].

In the present study, no patients reported post-surgery pain, proving that the administration of analgesic drugs as needed could be effective [33,34].

Post-surgical bleeding and wound infection were recorded in patients affected by type I diabetes more than patients of Group A, proving that vascular changes resulting from this disease could interfere with wound healing activities.

Kopman et al. [35] report that high blood glucose levels and non-enzymatic glycation of proteins lead to the formation of advanced glycation end products (AGEs), which could increase the permeability of the endothelium, increase the flux of inflammatory cells, and consequently cause micro-vascular complications and delayed wound healing [36,37].

A further factor that could influence all the parameters considered concerns the role of the patient’s hygienic maintenance and compliance with follow-up protocols, which could be significant for implant survival rate, marginal bone loss, prevention and early interception of post-operative and prosthetic complications [38,39,40].

## 5. Conclusions

Within the limitation of this study, the placement of dental implants supporting fixed prostheses could also be an option in patients with type I diabetes, provided that this is compensated and recent blood tests are examined by the clinician in advance of surgery.

In addition, subjecting the patient to hygiene maintenance therapy and periodic check-ups could prevent implant failure, higher-than-normal marginal bone loss due to bacterial inflammatory processes, and post-operative complications; if such problems develop, they could be intercepted early. Further clinical trial could be necessary to confirm the obtained results.

## Figures and Tables

**Table 1 ijerph-19-11735-t001:** Number of patients in each group classified according to the need for rehabilitation of the maxilla, mandible or both jaws and according to the type of rehabilitation performed (on six axial implants or on four in the All-on-Four protocol).

	Group A	Group B
N° patients	25	22
Need of maxilla rehabilitation	9	8
Need of mandible rehabilitation	7	6
Need of rehabilitation of both arches	9	8
N° of fixed rehabilitation on 6 axial implants	11	13
N° of fixed rehabilitation with All-on-Four protocol	14	9

**Table 2 ijerph-19-11735-t002:** Number of implants lost for each group and implant survival rate according to early and late failure.

	N° Implants	Early Failure	Late Failure	Implant Survival Rate
**Group A**	122	3	1	96.72%
**Group B**	114	4	2	94.74%

**Table 3 ijerph-19-11735-t003:** Average marginal bone loss for each group during the follow-up.

Marginal Bone Loss		
	Group A	Group B
6 months (mm)	0.57 ± 0.52	0.63 ± 0.54
1 year (mm)	0.99 ± 0.82	0.94 ± 0.75
2 years (mm)	0.84 ± 0.73	0.82 ± 0.64
3 years (mm)	0.86 ± 0.79	0.90 ± 0.76
4 years (mm)	1.00 ± 0.90	1.02 ± 1.00
5 years (mm)	1.02 ± 0.94	1.04 ± 0.87
6 years (mm)	1.03 ± 1.01	1.11 ± 0.68
7 years (mm)	1.04 ± 0.94	1.12 ± 0.78
8 years (mm)	1.06 ± 0.78	1.12 ± 0.89
9 years (mm)	1.07 ± 1.00	1.13 ± 0.91
10 years (mm)	1.09 ± 0.85	1.14 ± 0.88

**Table 4 ijerph-19-11735-t004:** Clinical complications classified by groups.

Clinical Complications	Group A	Group B
Edema	2	4
Pain	0	0
Bleeding	2	7
Wound infection	0	3

**Table 5 ijerph-19-11735-t005:** Prosthetic complications classified by groups.

Prosthetic Complications	Group A	Group B
Provisional prosthesis fracture	1	2
Provisional screw loosening (abutment)	3	2
Provisional screw loosening (prosthetic)	0	1
Detachment of the veneering material (final prosthesis)	0	0

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
