# Peer review of "Full Arch Implant-Prosthetic Rehabilitation in Patients with Type I Diabetes Mellitus: Retrospective Clinical Study with 10 Year Follow-Up"

_ijerph, 2022, doi:10.3390/ijerph191811735_

Round 1

Reviewer 1 Report

In the text it is unspecifically described that "Blood tests were prescribed about one month before surgery ".  It would be interesting to understand which lab tests were required.  Has the percentage of glycosylated hemoglobin been measured?  Have antigenic tests been performed (ICA, GADA, IA-2A, IAA) to see if the clinicians have also treated forms of type I autoimmune diabetes?

Author Response

Dear Reviewer,
thank you for your reply and request. 
The blood tests requested included evaluation of blood glucose, glycated haemoglobin and antigenic tests (ICA, GADA, IA-2A, IAA). 
We remain at your disposal for any further clarifications and thank you for your courtesy and helpfulness.

Reviewer 2 Report

The authors present a retrospective clinical study in which they assess and compare the implant survival, the marginal bone loss and clinical and prosthetic complications in healthy patients and patients with type I diabetes, undergoing full-arch implant-prosthetic rehabilitation. The authors present a well-edited and scientifically consistent research article, in a very relevant area in dentistry. Overall, the manuscript left me with a good impression, and it is an important study in the field of public health in implantology.

Although the study has clinical and scientific implications of enhancement, some corrections should be noted:

In a retrospective study, the formulation of the problem should be equated. It must be introduced a working hypothesis, that must be considered in the discussion and answered in the conclusion.

Author Response

Dear Reviewer, 
thank you for your examination and request,
in agreement with your consideration, we have added a specification in the aim of the study. 
We remain at your disposal for further clarification. 
